



# GNSS-based water vapor estimation and validation during the MOSAiC expedition

Benjamin Männel[1], Florian Zus[1], Galina Dick[1], Susanne Glaser[1], Maximilian Semmling[2], Kyriakos Balidakis[1], Jens Wickert[1,3], Marion Maturilli[4], Sandro Dahlke[4], and Harald Schuh[1,5]

[1]GFZ German Research Centre for Geosciences, Telegrafenberg, Potsdam, Germany
[2]DLR-SO Institute for Solar-Terrestrial Physics, Neustrelitz, Germany
[3]Technische Universität Berlin, Chair GNSS Remote Sensing, Navigation, and Positioning, Berlin, Germany
[4]Alfred-Wegener-Institut, Helmholtz-Zentrum für Polar- und Meeresforschung, Bremerhaven, Germany
[5]Technische Universität Berlin, Chair Satellite Geodesy, Berlin, Germany

**Correspondence:** Benjamin Männel (benjamin.maennel@gfz-potsdam.de)

**Abstract.** Within the transpolar drifting expedition MOSAiC (Multidisciplinary drifting Observatory for the Study of Arctic Climate), GNSS was used among other techniques to monitor variations in atmospheric water vapor. Based on 15 months of continuously tracked GNSS data including GPS, GLONASS, and Galileo, epoch-wise coordinates and hourly zenith total delays (ZTD) were determined using a kinematic precise point positioning (PPP) approach. The derived ZTD values agree

to $1.1\pm0.2$ mm (RMS of the differences 10.2 mm) with the numerical weather data of ECMWF's latest reanalysis, ERA5, computed for the derived ship's locations. This level of agreement is also confirmed by comparing the on-board estimates with ZTDs derived for terrestrial GNSS stations in Bremerhaven and Ny Ålesund and for the radio telescopes observing Very Long Baseline Interferometry in Ny Ålesund. Preliminary estimates of integrated water vapor derived from frequently launched radiosondes are used to assess the GNSS-derived integrated water vapor estimates. The overall difference of $0.08\pm0.04$ kg m$^{-2}$

(RMS of the differences 1.47 kg m$^{-2}$) demonstrates a good agreement between GNSS and radiosonde data. Finally, the water vapor variations associated with two warm air intrusion events in April 2020 are assessed.

## 1   Introduction

Troposphere delays are generally regarded as one of the primary error sources in GNSS positioning and therefore get modeled and estimated in the GNSS analysis process. By estimating these delays, it is possible to use GNSS for high-precise positioning

but also as a valuable tool to monitor the troposphere. Conventionally, the zenith total delay (ZTD) consists of two parts: the zenith hydrostatic delay (ZHD) and the zenith wet delay (ZWD). It is connected to the individual satellite observations and the associated slant delays by dedicated mapping functions and depends – apart from the air pressure-related (hydrostatic) part – on the partial water vapor pressure and therefore on the water vapor in the lower atmosphere (Elgered and Wickert, 2017). By estimating this wet part of the delay (i.e., the ZWD), GNSS observations allow to observe directly the amount of atmospheric

water wapor which is an active and most abundant component of the climate system (Alshawaf et al., 2018; Rinke et al., 2019). Atmospheric water vapor plays an essential role in today's climate variations, accounts for 60–70% of the greenhouse effect





(Kiehl and Trenberth, 1997; Guerova et al., 2016), and is an important input parameter for numerical weather models (NWM). In the generally dry Arctic, atmospheric moisture intrusions from lower latitudes affect the snow and sea ice cover by increased longwave radiation (Woods and Caballero, 2016). The estimation of ZTDs and the subsequent conversion into precipitable wa-

ter vapor (PWV) or integrated water vapor (IWV) is done operationally for many hundred land-based GNSS-stations and is confirmed to agree with conventional meteorological observations (e.g., Gendt et al., 2004; Shangguan et al., 2015; Steinke et al., 2015). According to Ning et al. (2016) the accuracy of GNSS-based IWV is at a level of 1-2 kg m$^{-2}$. However, with a few exceptions continuous and long-term GNSS-based water vapor time series over oceans are not available but are highly important for climate investigations. In the past, several authors investigated shipborne PWV retrieval and reported an agreement

at the 2 mm level compared to radiosondes (Fujita et al., 2008) and at the 3 mm level compared to radiometer data (Rocken et al., 2005). Boniface et al. (2012) investigated the ability to determine mesoscale moisture fields from shipborne GNSS data over four months. Wang et al. (2019) studied a 20-day ship cruise in the Fram Strait and reported an agreement for the PWV of 1.1 mm compared to weather models and radiosondes. Based on the four-months ship campaign, Shoji et al. (2017) reported the practical potential of kinematic precise point positioning (PPP) for water vapor monitoring over oceans worldwide with partic-

ular challenges during high-humidity conditions. Based on the large-scale EUREC$^4$A campaign, Bosser et al. (2020) presented IWV solutions derived from GNSS receivers on-board the research vessels RV Atalante, RV Maria S. Merian, and RV Meteor. Overall, they reported a good agreement with biases of $\pm 2$ kg m$^{-2}$ with respect to numerical weather models and terrestrial GNSS stations. For this study, we derived a 15 months zenith total delay and water vapor time series between summer 2019 and autumn 2020 observed by a GNSS receiver installed on-board the German research vessel RV Polarstern (Alfred Wegener

Institute, 2017) as part of the Multidisciplinary drifting Observatory for the Study of Arctic Climate (MOSAiC) expedition.

The main objective of the MOSAiC expedition was to investigate the complex climate processes of the Central Arctic for improving global climate models. RV Polarstern departed on September 20, 2019 in Tromsø, Norway and started the transpolar drift on October 4 at 85° N, 137° E. Interrupted for around four weeks due to a supply trip to Svalbard in May and June

2020, RV Polarstern ended the drift on August 9, 2020 at 79° N, 4° E. For the second part of the expedition, RV Polarstern returned to the Central Arctic in mid of August 2020 to observe the sea ice in its onset and early freezing phase. RV Polarstern finally returned to Bremerhaven on Oct. 12, 2020. The GNSS receiver was a continuously operational instrument within the ship-based atmosphere monitoring system. It was provided by the GFZ German Research Centre for Geosciences with the main motivation to derive water vapor variations continuously from ground and to allow a comparison for the radiosonde data.


Following this introduction, the GNSS installation on RV Polarstern and data availability is discussed in Sect. 2. The processing strategy applied in this study is summarized in Sect. 3 while Sect. 4 discusses the derived kinematic coordinates. In Sect. 5, the ZTD solution is assessed with respect to ERA5-based ZTDs and ZTDs derived from land-based GNSS and VLBI (Very Long Baseline Interferometry). Subsequently derived IWV values are discussed in Sect. 6 in comparison to preliminary

radiosonde data. The paper closes with a summary and conclusions in Sect. 7.





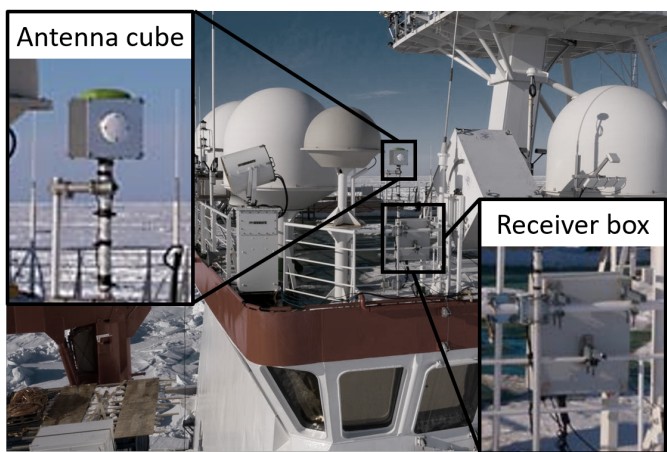

**Figure 1.** RV Polarstern's observation deck in May 2020; the GFZ GNSS antenna is mounted at portside; courtesy Torsten Sachs (GFZ).

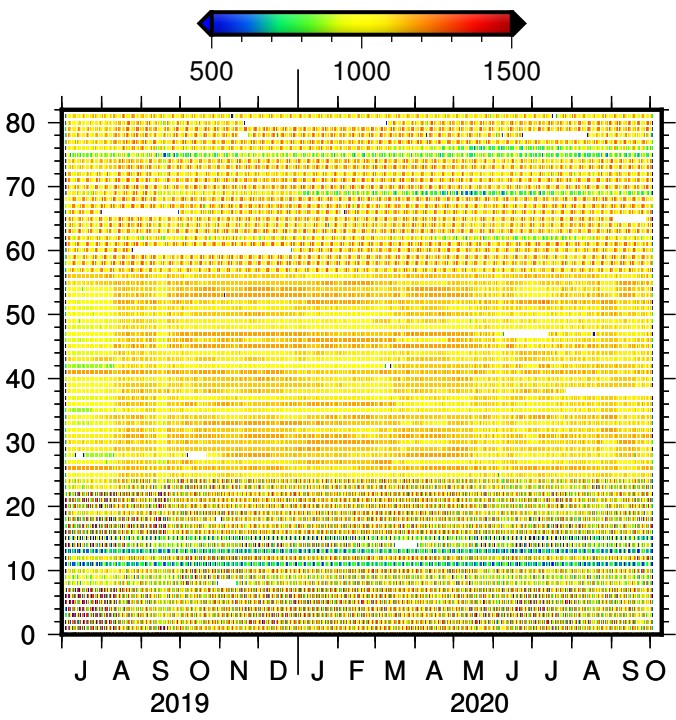

**Figure 2.** Number of L1 observations (phase of first frequency) per day and satellite; observation types are L1X for Galileo (rows 1-25), L1W for GPS (rows 26-57), L1P for GLONASS (rows 58-82) for GNSS receiver at RV Polarstern.



## 2 GNSS installation and data availablility

The GNSS equipment was installed on July 4, 2019 shortly after the end of RV Polarstern's previous expedition PS120 and at the beginning of a nearly six-week shipyard period. Consequently, GNSS data have been recorded during the stay at Bremerhaven, the expedition PS121 (Fram Strait), and the entire MOSAiC expedition (PS122). For logistical reasons, the receiver was switched off on October 3, 2020 at a position very close to Ny Ålesund, Svalbard. Therefore, we have 15 months of high-accuracy GNSS data which is very valuable for climate relevant studies.

Figure 1 shows GFZ's GNSS equipment installed at the RV Polarstern's observation deck. A JAV_GRANT-G3T antenna without a choke-ring was used. To support reflectometry, the antenna was mounted on a cube-structure together with a side-looking antenna (Semmling et al., 2021). The receiver equipment (geodetic JAVAD_TR_G3TH receiver) was stored in a cabinet mounted at the observation deck's railings. As visible in Fig. 1, the antenna location is not perfect as being subject to shadowing and strong multipath effects caused by the nearby radomes and the crow's nest. Due to limited data bandwidth during the cruise data transfer in real-time was not possible. Data post-processing started after the cruise at GFZ. The raw data were converted using the JPS2RIN converter software (version v.2.0.191). The derived RINEX files were spliced, sampled, and checked using gfzrnx (Nischan, 2016).

The considerable large number of received observations, represented in Fig. 2 in terms of L1 phase observations, is first of all promising. With a sampling rate of 30 s, around 1000 L1 phase observations were tracked per satellite and day over the complete time span of 15 months. Visible constellation specific patterns are expected as they are caused by the satellite orbit configurations, i.e., inclination, repeat cycle, and revolution period.

## 3 Processing strategy

The Bernese GNSS Software 5.2 (Dach et al., 2015) was used for the data processing, which was performed as kinematic PPP (Zumberge et al., 1997) with epoch-wise estimated coordinates and hourly estimated zenith total delays among other parameters. The kinematic approach is needed to account for the traveling periods, height variations due to tides and waves, and the drift phases which showed an average speed of 12 km per day (corresponding to 4 m within the observation interval of 30 s). For validation purposes, the shipyard period in Bremerhaven (i.e., the dry dock period) was processed consistently in kinematic mode. The resulting kinematic coordinates are studied in Sect. 4. Overall, 25 ZTD values are estimated per day together with 8640 kinematic coordinates, 2880 clock corrections, around 80 differential code biases, and around 280 ambiguities. The piecewise-linear ZTD estimates are constrained relatively with 1 mm. Table 1 provides a summary of the modeling and parametrization strategy. According to the capabilities of the processing software, ambiguity fixing was not applied, therefore, ambiguity parameters are estimated. Driven by the receiver's high latitude and the free horizon on the portside a low elevation cutoff angle of 3° was chosen. An elevation-dependent observation weighting using $\cos^2(z)$ was applied with $z$ as zenith angle.


**Table 1.** Summary of estimation and processing strategy.

| modeling and a-priori information | |
| --- | --- |
| observations | ionosphere free-linear combination formed by undifferenced GPS, GLONASS, and Galileo observations, sampled with 30s, elevation cutoff 3°, elevation-dependent weighting using $\cos^2(z)$ |
| a priori products | CODE MGEX orbits, clock corrections, Earth rotation parameters (Prange et al., 2020) |
| tropospheric correction | hydrostatic delay computed based on VMF, mapped with VMF (Böhm et al., 2006) |
| ionospheric correction | 1st order effect considered with ionosphere-free linear combination |
| GNSS phase center | igs14_2129.atx (Rebischung and Schmid, 2016) |

| estimated parameters | |
| --- | --- |
| kinematic coordinates | pre-eliminated every epoch |
| troposphere | 25 zenith delays per day; constrained relatively with 1 mm; mapped with VMF |
| receiver clock | pre-eliminated every epoch |
| GNSS ambiguities | estimated, without ambiguity fixing |
| differencial code biases | per satellite |

Some general processing indicators are highlighted in Fig. 3. On average, 40'000 to 60'000 ionosphere-free observations remained after pre-cleaning for the daily processing, which is equivalent to 13–20 observations per epoch. As shown in the middle panel of Fig. 3, singular epochs, i.e., epochs with fewer than four observations, occurred for a number of days, mainly between December 2019 and March 2020. Due to the high latitude of the ship's position, satellite observations with very low elevations dominate during this period. Further signal delays are expected due to ice accumulation on the antenna and increased multipath effects caused by the surrounding instruments. The lower panel of Fig. 3 shows the MP12 multipath values derived from the RINEX files using TEQC (Estey and Meertens, 1999). Due to the antenna surrounding, large multipath effects of around 1.5 m occurred. According to Bosser et al. (2020) this value of 1.5 m is large compared to the multipath observed at RV Atalante und RV Meteor (in both cases, the GNSS antenna was installed at the crow's nest), but smaller than the multipath derived for RV Maria S. Merian, where the GNSS antenna was placed also on the observation deck. Between January and March 2020, the multipath increased by around 6 % compared to the previous three months. As visible in Fig. 3, a larger number of singular epochs occurred also during the port departures on August 10, 2019 (Bremerhaven, 381 epochs) and September 20, 2019 (Tromsø, 495 epochs), interestingly this is not the case for the arrival in Tromsø. During the "refueling & personnel rotation 4" (August 10-12, 2020) where the Akademik Tryoshnikov was moored alongside RV Polarstern, the multipath parameter raised to values above 2 m, and the number of processed observations dropped below 30'000. The multipath strongly increased caused by additional reflections, especially as the Akademik Tryoshnikov moored at RV Polarstern's port side. The number of singular epochs increased to 182, 216, and 129 epochs for the three days. Overall, these pre-processing results demonstrate and ensure precise and reliable results.





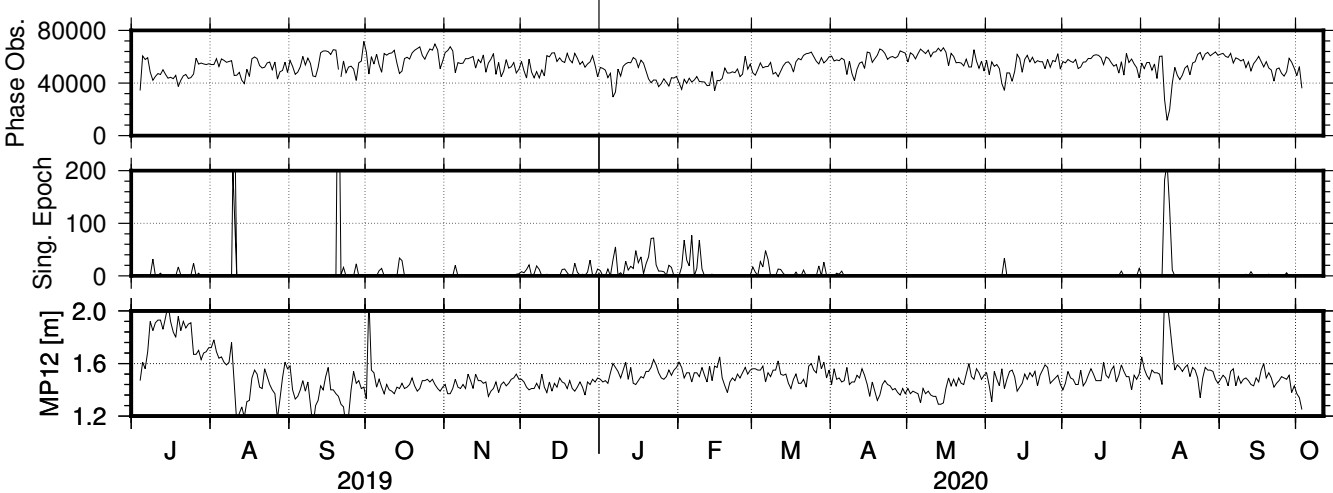

**Figure 3.** Number of phase observations per day used in the final adjustment (*upper panel*), number of singular epochs per day (i.e., epochs with fewer than four observations, based on 30-sec sampling, *middle panel*), and daily TEQC-based multipath MP12 values (*lower panel*).

## 4  Assessment of kinematic coordinates

In general, kinematic coordinates have to be estimated for a shipborne GNSS receiver to account for the ship's motion. Compared to static coordinates, kinematic coordinates cannot be assessed easily due to missing repetitions of positions (unlike the repeatability check for permanent stations) and the usual absence of any ground-truth information as for example, known marker coordinates. Therefore, the assessment of kinematic coordinates is possible only during specific periods.

One specific period was during the shipyard stay in Bremerhaven, where RV Polarstern spent nearly four weeks in the dry dock and the antenna position could be assumed to be static and thus precisely assessed. Fig. 4 shows the coordinate variations between July 7 and August 3, 2019 for North, East, and Up direction. For the horizontal components, 79 % and 83 % of the coordinates are within ±4 cm. A larger variation is expected for the height with 78 % of the coordinates being within ±8 cm. The standard deviation of the derived horizontal coordinates is 4.8 and 5.7 cm for North and East, respectively. However, it has to be noted that during the shipyard stay multipath increased while the number of observations decreased significantly (see Fig. 3). This is most probably related to additional obstacles and reflections due to construction work. A mean ellipsoidal height of 61.49±0.01 m was determined. Using a geoid height of 39.58 m at Bremerhaven this corresponds to an antenna height of 21.91 m during the dry dock phases.

The second period for a coordinate validation is the harbor stay in Tromsø, Norway. Without a ship motion the ocean tides can be used as ground truth, thus, the correlation between observed height and tidal record provides a validation opportunity. Fig. 5 shows the GNSS-based height variations for September 12 to 20, 2019. A nearly perfect agreement compared to a close-by

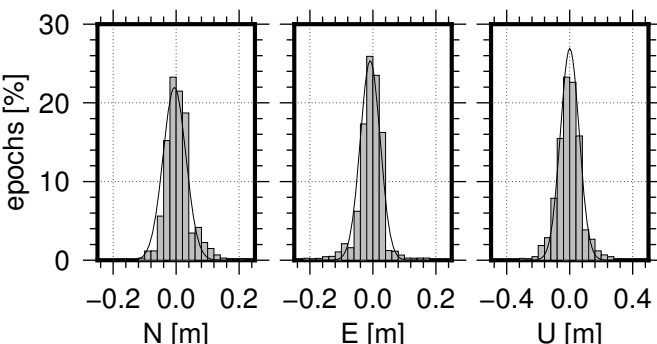

**Figure 4.** Histogram of coordinate variations in North, East, and Up direction for the dry dock period of RV Polarstern (July 7 to August 3, 2019), a mean coordinate was subtracted; please note the different scales for horizontal and vertical components.

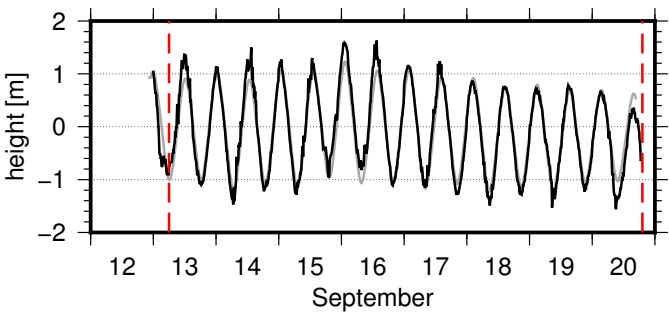

**Figure 5.** Estimated height coordinates during the stay at Tromsø (*black*); the Tromsø tide gauge record is plotted for comparison (*grey*), an empirically estimated height difference of 22.2 m was subtracted from the GNSS positions; vertical red lines indicate RV Polarstern's arrival and departure times in Tromsø.

tidal record[1] is visible with a correlation coefficient of 0.97. An empirically estimated height difference of 22.2 m is subtracted from the GNSS positions.

Overall, kinematic coordinates for nearly 1.3 Mio. epochs were estimated from kinematic PPP. Whereas 0.5 % of all epochs are singular (i.e., epochs with less than four satellites) and another 0.9 % are interpolated but not estimated (see also Fig. 3) ensuring the high-quality of the data.

## 5    Assessment of zenith total delay

The derived ZTDs are analyzed and the results are presented in this section. The assessment and validation process includes the comparison to numerical weather model data (Sect. 5.1), a comparison against ZTD derived for selected onshore GNSS

---

[1]Tide gauge data are taken from http://vannstand.no/en/sehavniva/Lokasjonsside/?cityid=9000020&city=Troms%C3%B8, assessed November 2020





**Table 2.** List of restricted periods for which atmospheric measurements were not permitted.

|   | Year | Entry | Exit | Remark |
|---|------|-------|------|--------|
| A | 2019 | Sep 12 00:00 | Sep 26 12:40 | economic exclusive zones Norway and Russia |
| B | 2020 | Jun 03 20:36 | Jun 08 20:00 | territorial waters Svalbard |
| C | 2020 | Oct 02 04:00 | Oct 02 20:00 | territorial waters Svalbard |
| D | 2020 | Oct 03 03:15 | Oct 04 17:00 | territorial waters Svalbard |

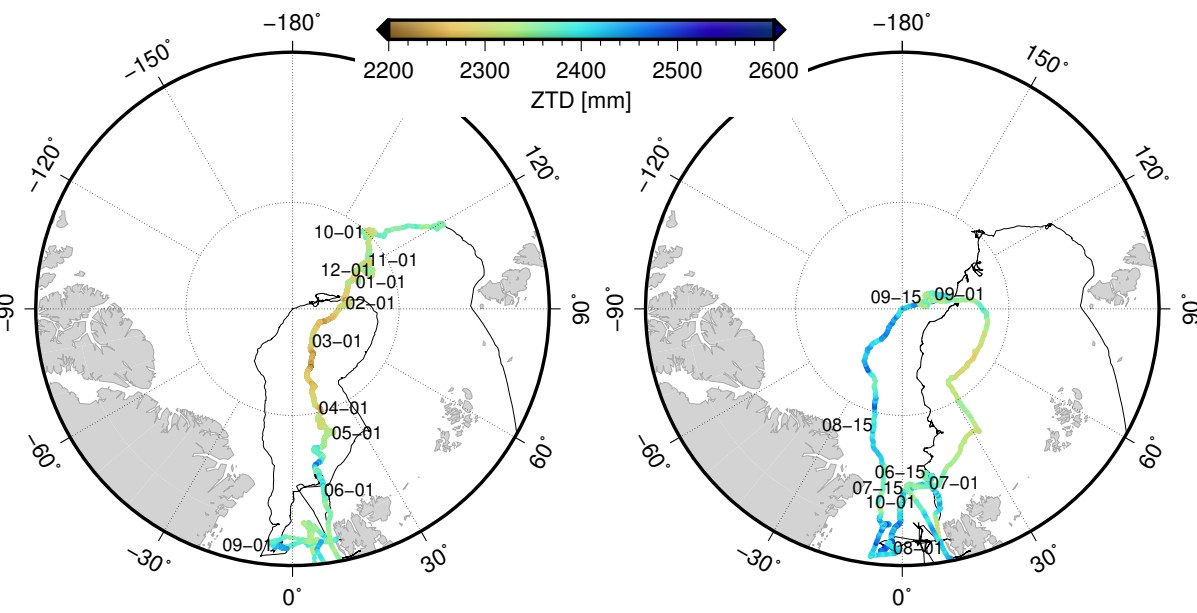

**Figure 6.** Ship track with hourly ZTD values (*color-coded according to the ZTD*); the left panel shows the ZTD series for August 2019 to June 05, 2020, the right panel shows the ZTDs for June 06 to October 03, 2019; selected time stamps are added.

135    stations (Sect. 5.2), and to ZTD derived by VLBI at the radio telescopes in Ny Ålesund (Sect. 5.3). During the entire 457 days, 10973 unique ZTD values are estimated, i.e., 24 per day, while the 25th is computed for the first epoch of the following day. In accordance to the MOSAiC data guidelines and research agreements, water vapor results are restricted during (1) the harbor stay in Tromsø, (2) the subsequent passage of the economic exclusive zone of Norway and Russia, and (3) all periods within the territorial waters (12 nautical miles) around Svalbard. Table 2 summarizes the affected periods. Following

140    these restrictions, the number of investigated ZTDs reduces to 10'503. Due to singular epochs 54 ZTDs and due to interpolated epochs another 91 ZTDs (corresponding to 0.5 % and 0.8 % of all ZTDs) are excluded from the following statistics. In addition, ZTDs estimated for epochs with fewer than 800 observations are excluded (192 epochs, corresponding to 1.8 %). Overall, 10'166 GNSS-based ZTDs (96.8 %) are considered in the following comparisons. Related to the applied constraint, the formal errors of the remaining ZTDs are relatively small with 99.6 % being below 4 mm.





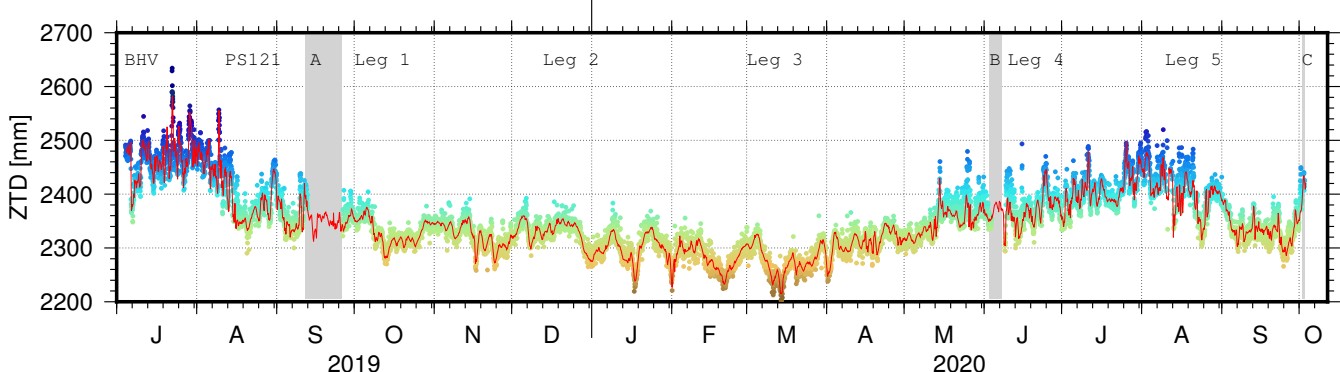

**Figure 7.** ZTD time series: hourly ZTD values *(color-coded according to the ZTD, same scale as in Fig. 6)* and 3-hourly ERA5-based ZTDs *(red line)*; the cruise parts are indicated by labels, restricted time periods are grey-shaded.

## 5.1  ZTD time series and comparison to ERA5

Figure 6 shows the ship track of RV Polarstern between August 2019 and October 2020. For clarity the figure is split into two panels showing the time August 2019 to June 5, 2020 (left) and June 6 to October 3, 2020. The left panel shows the ZTD variations during the Fram strait expedition and the transpolar drift until the "resupply & personnel rotation 3" in June 2020. Very low ZTD values are observed especially during the polar night which lasted from mid October till mid of March. The right panel shows the ZTD during the second part of MOSAiC with another drift phase, reaching the North Pole, and crossing the Central Arctic in autumn 2020. Figure 7 presents the hourly ZTD time series derived for the entire period, including the shipyard stay at Bremerhaven, the Fram Strait expedition PS121, and the MOSAiC expedition. According to the ZTD values, three periods could be identified: (1) relatively wet phase (ZTD > 2500 mm over the entire day) during July to August of both years (i.e., 2019 in Bremerhaven and 2020 in the Central Arctic), (2) periods with ZTD entirely below 2400 mm during the transpolar drift until May 2020, and (3) periods with ZTD between 2300 and 2500 mm in the transition phases. For comparison, 3-hourly ZTD values derived from the European Centre for Medium-Range Weather Forecasts (ECMWF) Re-Analysis 5 (ERA5) (Hersbach et al., 2020) are added to Fig. 7. The method described in Zus et al. (2012) is utilized to calculate the ZTDs in the weather model analysis. Overall, a difference of 1.1±0.2 mm with an RMS of 10.2 mm could be found between the ship-based GNSS and ERA5 ZTDs, showing that GNSS ZTDs are slightly larger than predicted by ERA5. For the much shorter Fram Strait expedition of RV Lance in August and September 2016, Wang et al. (2019) reported a better agreement of overall 0.8 mm and an RMS of 6.5 mm. For the MOSAiC dataset, we applied an outlier detection based on a 3.0-$\sigma$ criteria (i.e., three times the standard deviation). Overall, 8.4 % of the differences are excluded.

Fig. 8 shows the direct comparison between the GNSS and the ERA5-based ZTD values. The correlation coefficient between both time series reaches 0.97, which agrees very well with the 97.2 % correlation presented in Wang et al. (2019). Interestingly,





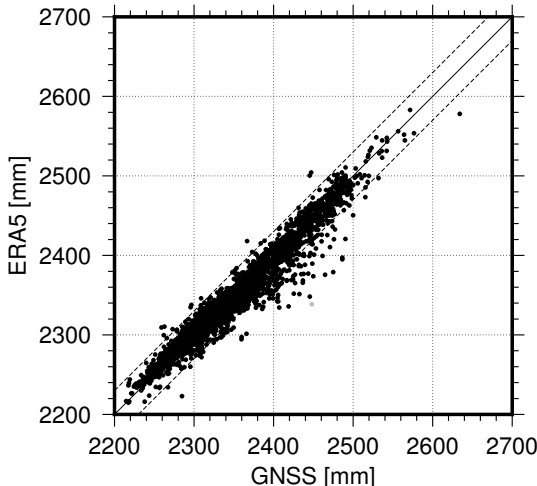

**Figure 8.** Comparison of ZTD values from shipborne GNSS and ERA5.

epochs where GNSS-based ZTDs are larger than the ERA5-based value occurred predominantly during summer months in 2019 and 2020.

## 5.2 Comparison to onshore GNSS

The assessment of ZTDs determined for the GNSS receiver on-board RV Polarstern with respect to ZTDs derived for onshore
GNSS receivers allows a second comparison option. A comparison between ship-based ZTD and land-based GNSS products is in general possible for (1) harbor stays with a close-by GNSS station and (2) periods where the ship's distance to a terrestrial reference station does not exceed a few hundred kilometers under stable weather conditions. The first comparison approach can be applied for the shipyard stay in Bremerhaven (six weeks). Unfortunately, the harbor stay at Tromsø (one week) could not be used for a comparison due to the data restrictions. The second approach is challenging given the remote character of
the MOSAiC expedition. However, during PS121, the re-supply trip at Svalbard and RV Polarstern's return trip, the distance between the GNSS tracking stations at Ny Ålesund, Svalbard and RV Polarstern was shorter than 200 km for several days. The related ZTDs are thus compared to the onshore GNSS at Ny Ålesund. Reference ZTDs have been estimated within operational GNSS processing system for meteorological applications, which is based on the GFZ Earth Parameter and Orbit determination System (EPOS.P8) software (Gendt et al., 2004; Wickert et al., 2020).


For the shipyard stay in Bremerhaven, a reasonable mean difference of $1.5\pm0.4$ mm and an RMS of 9.9 mm was estimated between ZTDs for RV Polarstern and the German SAPOS station 0994. This station is located approx. 2.6 km from RV Polarstern and around 20 m above the ground like the GNSS antenna height at Polarstern. Therefore, no height correction was applied. Figure 9 shows the time series of derived ZTDs during the harbor and shipyard stay in Bremerhaven, including corre-
sponding ZTD estimates for the SAPOS station 0994, and the ERA5-based ZTD time series as additional reference. Overall, a





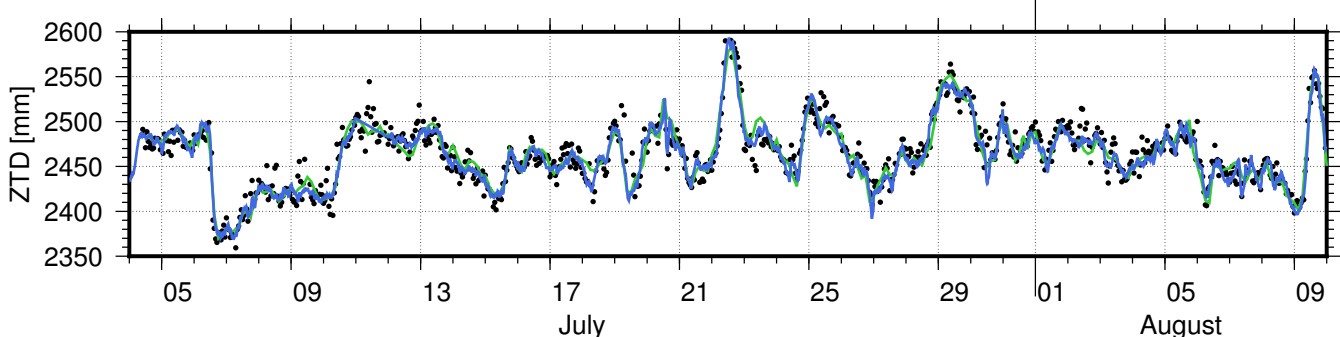

**Figure 9.** ZTD time series during the shipyard stay in July and August 2019: hourly ZTD values from RV Polarstern (*black dots*) and SAPOS station 0994 (*blue*), as well as ERA5-based ZTDs (*green*).

**Table 3.** Summary of differences with respect to GNSS and ERA5 (always computed as difference RV Polarstern − reference); $\Delta_h$ horizontal distance, $\Delta_v$ vertical difference; number of used / all samples is given; TTW = territorial waters; units: mm.

| Location | Reference | Period | Offset | RMS | Samples | Remarks |
|---|---|---|---|---|---|---|
| Bremerhaven | 0994 | 2019 Jul 4 - Aug 10 | 1.5±0.4 | 9.9 | 708 / 807 | $\Delta_v$ not corrected |
| $\Delta_h = 2.6$ km | ERA5 | 2019 Jul 4 - Aug 10 | 0.4±0.7 | 12.0 | 277 / 287 | |
| Ny Ålesund | NYA2 | 2019 Aug 13 - Sep 11 | 6.7±0.7 | 14.2 | 420 / 461 | $\Delta_v = 27$ m corrected |
| up to $\Delta_h = 200$ km | ERA5 | 2019 Aug 13 - Sep 11 | 6.9±0.9 | 13.5 | 216 / 232 | |
| | NYA2 | 2020 Jun 2-6 & Jun 9 | 5.0±1.7 | 9.3 | 28 / 28 | $\Delta_v = 27$ m corrected, outside TTW |
| | ERA5 | 2020 Jun 2-6 & Jun 9 | 2.2±3.0 | 11.8 | 15 / 15 | |
| | NYA2 | 2020 Oct 2 - 3 | 10.1±2.6 | 9.3 | 13 / 13 | $\Delta_v = 27$ m corrected, outside TTW |
| | ERA5 | 2020 Oct 2 - 3 | -1.5±2.2 | 4.9 | 5 / 5 | |

good agreement can be noted between the three ZTD solutions while the sparser sampling of the ERA5 ZTD solution is visible (i. e., 3h sampling for ERA5 compared to 1h for GNSS). During this time an RMS of 12.0 mm and an average of 0.4±0.7 mm is observed for the differences to the ERA5-based ZTDs. More details are provided in Tab. 3.

For Tromsø only an indirect comparison between the ERA5-based ZTD time series and the permanent GNSS station TRO1 is possible. TRO1 is a GNSS station provided via the International GNSS Service (IGS, Johnston et al., 2017) and is located approx. 3 km from RV Polarstern's mooring and around 107 m above sea level. Consequently, these ZTDs were corrected for a height differences of 85 m using a rough delay correction of $0.3\,\mathrm{mm\,m^{-1}}$ for the hydrostatic part. A difference of 1.0±0.8 mm and an RMS of 6.1 mm reveals a good agreement for this comparison between GNSS and ERA5.




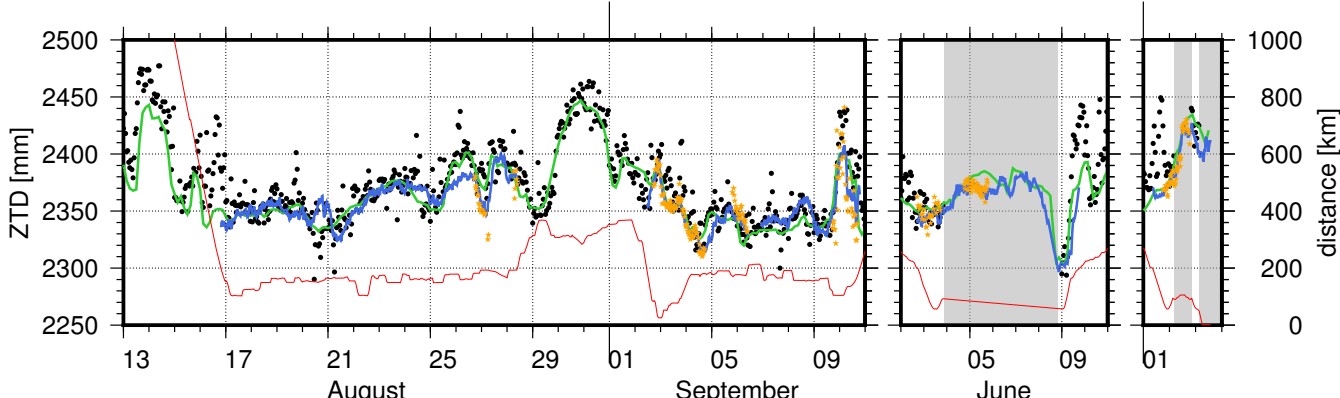

**Figure 10.** Hourly ZTD values (*black*) compared to ZTDs derived for the IGS station NYA2 (Ny Ålesund, Svalbard, *blue*) during August /
September 2019 (*left*), June 2020 (*middle*), and October 2020 (*right*) and to ZTDs derived from ERA5 (*green*) and VLBI stations NYALES20
and NYALE13S (*orange*); the distance between RV Polarstern and NYA2 is represented in *red*.

The comparison between the ZTDs derived for RV Polarstern and the GNSS station in Ny Ålesund, Svalbard is more chal-
lenging due to the larger distances, the ship's speed, and partly the performed ship operations. In addition, also the orographic
setting might be a source for differences between the RV Polarstern measurements on open water and the measurements at
the edge of a fjord surrounded by mountains in Ny Ålesund. The reference station (NYA2) is a GNSS station operated by
GFZ (Ramatschi et al., 2019) and observations and metadata are available within the IGS. Figure 10 shows Polarstern's ZTD
series and the ERA5-based ZTD values. In addition, the geometrical distance between RV Polarstern and NYA2 is indicated
by the red line. Whereas the observed good agreement between the on-board estimates and ERA5 is expected, the differences
regarding NYA2 are partly larger. Especially during the PS121 expedition (Fram Strait), time shifts between the ZTD series
can be observed for some periods, e.g., for August, 26–28. For these particular dates the PS121 expedition report mentions
a storm field close to Iceland affecting RV Polarstern with speeds of about 8 Bft (Metfies, 2020). During the re-supply stay
in June 2020, ZTDs are not permitted. However, less accurate ZTDs are expected for this period considering the drop in the
processed phase observations (see Fig. 3) potentially caused by the logistic activities. For the arrival and departure periods the
agreement is within the expected range. For the third interval, again, a good agreement is visible for October 2 and 3, 2020 but
larger differences for the approaching period on October 1, 2020. While RV Polarstern approached Ny Ålesund closely with
distances to NYA2 below 2 km, the corresponding values are, however, restricted by the research agreement and cannot be used
for the comparison. The statistics are summarized in Tab. 3. Overall, offsets and RMS are below 10 and 14 mm, respectively.
Comparing to Wang et al. (2019) and Bosser et al. (2020), larger variations are derived for RV Polarstern most probably caused
by the sub-optimal antenna position.



**Table 4.** Summary of differences with respect to VLBI (always computed as difference RV Polarstern − reference); number of used / all samples is given; units: mm.

| Location | Reference | Period | Offset | RMS | Samples | Remarks |
|----------|-----------|--------|--------|-----|---------|---------|
| Ny Ålesund | NYALES20 | 2019 Aug 26 - Sep 11 | 9.3±2.2 | 16.5 | 54 / 66 | $\Delta_v = 33$ m corrected |
| up to $\Delta_h = 200$ km | NYALE13S | 2020 Jun 2 - 9 | 3.0±2.5 | 12.7 | 25 / 25 | $\Delta_v = -1$ m corrected, outside TTW |
| | NYALES20 | 2020 Oct 2 - 3 | 13.6±4.6 | 12.2 | 7 / 7 | $\Delta_v = 33$ m corrected, outside TTW |

## 5.3 Comparison to VLBI

The geodetic fundamental site in Ny Ålesund also allows a comparison between the ZTDs determined for the GNSS antenna on-board RV Polarstern and the ZTD observed by VLBI at the radio telescopes NYALES20 and NYALE13S[2] for a completely external validation. Very Long Baseline Interferometry is an interferometric technique measuring the time delay between the reception of signals transmitted by extragalactic radio sources at two or more antennas (Schuh and Behrend, 2012). Currently, VLBI sessions with global networks are not performed continuously but scheduled in twice weekly 24h sessions and provided

within the International VLBI Service for Geodesy and Astrometry (IVS, Nothnagel et al., 2017). The high accuracy of VLBI-based troposphere estimates has been reported for example by Heinkelmann et al. (2007) and Balidakis et al. (2018). In the time span August 11 till September 12, 2019 in total nine 24h sessions with NYALES20, during May 20 and June 15, 2020 seven sessions with NYALES20 or NYALE13S, and during September 20 and October 4, 2020 another five sessions with NYALES20 were analyzed using PORT (Potsdam Open Source Radio Interferometry Tool). PORT is GFZ's VLBI analysis

software and is based on VieVS (Vienna VLBI Software, Böhm et al., 2012, Nilsson et al., 2015). The derived VLBI-based ZTDs are shown in orange in Fig. 10. First of all, it can be noted that the ZTDs of GNSS (NYA2) and VLBI agree well, which is expected due to the short horizontal distances between the stations[3] and the highly accurate space techniques GNSS and VLBI. Overall, a good agreement is visible also between the VLBI and the RV Polarstern ZTDs. However, a few VLBI-based ZTDs differ significantly in August 2019, while also the GNSS ZTDs showed some larger differences for these days as reported

above. For June and October 2020, there were VLBI sessions each while RV Polarstern was close to Svalbard. However, during these periods RV Polarstern was mainly within the territorial waters around Svalbard in which ZTDs are restricted. In June, one session with NYALES20 is within the restricted period, while a comparison is allowed for a session with NYALE13S. For this session an offset of 3.0 mm and an RMS of 12.7 mm is determined over 25 ZTD differences. For October a comparison is possible as well, however, only for a short period of seven hours. Similar to the comparison against onshore GNSS, offsets and

RMS values are below 16 mm. The statistical values are summarized in Tab. 4.

---

[2]active since Jan 8, 2020

[3]273 m for NYA2–NYALES20 and and 1539 m for NYA2–NYALE13S



## 6 Assessment of integrated water vapor

This section discusses integrated water vapor values derived from the ZTD on-board RV Polarstern. The conversion between ZTD and IWV was performed applying Eq. 2 described in Bevis et al. (1994). The zenith wet delay was computed by subtracting the hydrostatic delay provided by ERA5 from the estimated ZTD values. The weighted mean temperature of the atmosphere $T_m$ was calculated from the ERA5 data using Eq. A18 given in Davis et al. (1985). To derive hourly IWV the 3-hourly ERA5 data are linearly interpolated.

From board RV Polarstern, radiosondes were launched every six hours during the entire MOSAiC expedition, and moreover every three hours for periods of specific interest. Based on the relative humidity data in the preliminary radiosonde dataset (Maturilli et al., 2021), vertically resolved specific humidity profiles were calculated applying Hyland and Wexler (1983) and integrated over the atmospheric column to retrieve IWV. Measurements for which the radiosondes did not reach a height of at least 10'000 m are excluded in the following (0.9 %).

Figure 11 shows the comparison between the GNSS-based IWV and the radiosonde observations. Overall an agreement of $0.08\pm0.04\,\mathrm{kg\,m^{-2}}$ with an RMS of $1.47\,\mathrm{kg\,m^{-2}}$ can be found together with a correlation coefficient of 0.97 between both datasets. In this comparison, 2.6 % of the overall 1495 difference, i.e., those exceeding $5\,\mathrm{kg\,m^{-2}}$, are excluded. Comparable values are reported by Shoji et al. (2017) for a comparison between radiosonde-based PWV and GNSS in the Northern Pacific. Bosser et al. (2020) reported slightly larger IWV biases with respect to ERA5 but similar variations with 2.2 and $2.7\,\mathrm{kg\,m^{-2}}$ for RV Atalante and RV Meteor, respectively. The majority of the absolute IWV values is below $5\,\mathrm{kg\,m^{-2}}$ as visible in Fig. 11. This result could be expected as driven by the low air temperatures, the amount of atmospheric water vapor was very low during large parts of the transpolar drift. Consequently, IWV values observed by GNSS and radiosondes are below $5\,\mathrm{kg\,m^{-2}}$ from mid of October 2019 till end of April 2020 with only a few exceptions. One example for such rapid moisture increase occurred in April 2020 associated with two warm air intrusion events on April 16 and 19. According to Magnusson et al. (2020), the warm air was pushed to the northeast in front of a low pressure trough over Scandinavia in the first event. In contrast, the second event was driven by warm air transported northward on the western side of a high pressure ridge that developed over Scandinavia. Both events on April 16 and 19 are well visible in the IWV time series shown in Fig. 12. For both events, the air temperature increased rapidly from around -20°C to nearly 0°C. Simultaneously the IWV observed by GNSS increased from below $5\,\mathrm{kg\,m^{-2}}$ to 8 and $13\,\mathrm{kg\,m^{-2}}$ for the two events. For both events, a nearly perfect agreement between GNSS-derived and radiosonde-based IWVs is visible.

## 7 Conclusions

The MOSAiC expedition offered a unique opportunity to study polar environmental conditions during one full annual cycle. Besides other techniques, an on-board GNSS receiver allowed to monitor the variations of atmospheric water vapor above RV Polarstern. Based on 15 months of continuously tracked GNSS data, a kinematic PPP approach including GPS, GLONASS,



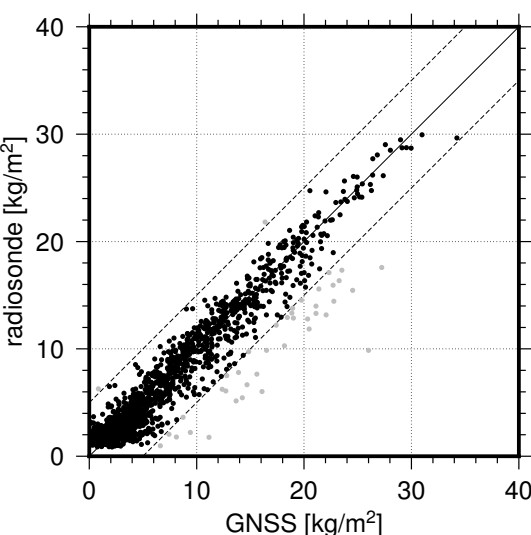

**Figure 11.** Comparison of IWV derived from GNSS and radiosondes; differences $>5\,\mathrm{kg\,m^{-2}}$ are shown in gray (2.6 % of in total 1495 differences).

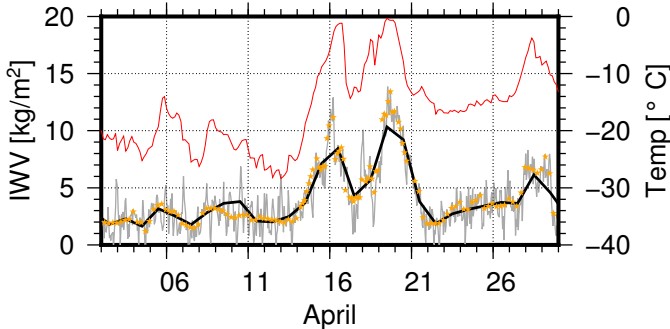

**Figure 12.** GNSS-based IWV: hourly values (*grey*) and daily averaged (*black*); Radiosonde-based IWV (*orange stars*); air temperature from ERA5 (*red*)

.





and Galileo was used to determine epoch-wise coordinates and hourly zenith total delays. By assessing the GNSS data itself,
a reliable number of observations was found, however, disturbed by multipath effects due to sub-optimal antenna location.
With a few exceptions, the kinematic coordinates are well determined over the entire time span. For the static shipyard stay the
variations of the kinematic coordinates were within 5 cm for the horizontal and within 10 cm for the vertical component. The
comparison of the GNSS-based ZTDs against ZTD derived from the ERA5 model shows a good agreement with an offset of
$1.1\pm0.2$ mm and an RMS of 10.2 mm over the entire period and a strong correlation of 0.97. Due to the remote character of the
MOSAiC expedition the comparison to terrestrial GNSS receivers was more challenging. For the harbor stay in Bremerhaven
we derived an offset of $1.5\pm0.4$ mm and a variation of around one centimeter. Comparing ZTDs over up to 200 km against
the IGS station NYA2, larger biases of up to 10 mm and standard deviations up to 14 mm were noted and confirmed by
comparing to ZTDs measured at the VLBI radio telescopes in Ny Ålesund. Thanks to frequent radiosonde measurements
during the MOSAiC expedition a detailed comparison between GNSS-based IWV and radiosonde measurements was possible.
The overall difference of $0.08\pm0.04$ kg m$^{-2}$ and the RMS of 1.47 kg m$^{-2}$ show a good agreement of both techniques which is
also visible during two warm air intrusions in April 2020. Overall, GNSS receivers on-board ships allow a cost-efficient and
continuously monitoring of atmospheric water vapor over the oceans.

*Data availability.*   GNSS RINEX data and derived ZTD and IWV values are available at Männel et al. (2021) and Männel and Zus (2021),
respectively.

*Author contributions.*   B.M., F.Z. and G.D. defined the study with the support from J.W., the PI of the GNSS experiment from GFZ during
MOSAiC. B.M., F.Z., S.G. and K. B. processed the GNSS, ERA5, and VLBI data, M.M. and S.D. provided the radiosonde data. All authors
contributed to the analysis, interpretation, and discussion of the results. B.M. prepared the manuscript with major contributions from F.Z.,
S.G. and M.M. All authors read and approved the final manuscript.

*Competing interests.*   The authors declare that they have no conflict of interest.

*Acknowledgements.*   The authors want to thank IGS, IVS, CODE, and ECMWF for making the used data and products publicly available.
We would also like to thank Thomas Gerber, Sylvia Magnussen, Markus Bradke, and the technical staff of RV Polarstern for all support
during GNSS installation, operations, and data retrieval. Radiosonde data were obtained through a partnership between the leading Alfred
Wegener Institute (AWI), the Atmospheric Radiation Measurement (ARM) User Facility, a U.S. Department of Energy facility managed by
the Biological and Environmental Research Program, and the German Weather Service (DWD). Data used in this manuscript was produced
as part of POLARSTERN cruise AWI_PS121 and of the international Multidisciplinary drifting Observatory for the Study of the Arctic
Climate (MOSAiC) with the tag MOSAiC20192020 (AWI_PS122_00).





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
