# Peer review of "GNSS-based water vapor estimation and validation during the MOSAiC expedition"

_Atmospheric Measurement Techniques, 2021_

## Author Response (AR1)

Reply on "GNSS-based water vapor estimation and validation during the MOSAiC expedition" by Benjamin Männel et al., Atmos. Meas. Tech. Discuss., https://doi.org/10.5194/amt-2021-79-RC2, 2021

Our answers are indicated by „A:"

Referee #1

This is an article on our present day ability to derive good quality ZTDs from GNSS data obtained on a moving platform, in this case a slow moving ship. It is interesting, wellwritten, easy to read, and deserves publication. In principle the manuscript can be published as is, but I have a few suggestions for improvements for the authors to consider.

> A: Thank you for the encouraging review and your comments.

1 The term "crows nest" will to most readers mean something else. Consider to just remove the first entry and say "top of ship" or "top of mast" in the second.

> A: Actually "crows nest" is the lookout point (like a cabin) at the mast. You are right it might be misleading to some readers. We added a short explanation when using "crows nest" for the first time.

2 Consider to include statistics on ZTDs from NYA2 versus ERA, possibly also from NE Greenland if you have easy access to GNSS data from there. Those from NYA2 you have already.

> A: We considered the usage of additional stations while defining the study. However, Polarstern's trajectory was not close enough to use GNSS stations in Greenland in a reliable way. Especially in Kronprins Christian Land which is closest to Polarstern's trajectory we are not aware of any permanent GNSS station (in addition to the tracking networks of IGS, G-NET and UNAVCO (e.g. https://www.unavco.org/data/dai/?lat=72.131764&lon=1.400925&zoom=2.73).

3 The processing is done after the expedition. Include a few sentences whether the quality of the GNSS ZTDs would be different was it done in near real-time, which is important for the potential use of ZTDs from ships in NWP.

> A: The quality in near real-time would not differ much (for static stations described e.g. in Gendt et al., 2004), however, given the very limited bandwidth it was not possible to transfer the GNSS data in near real-time. Processing on the ship would be possible but was not done as the focus was on collecting the measurements. Nevertheless it might be a very interesting approach for further developments.

4 Ground based ZTDs are (to my knowledge) not assimilated in ERA, which strengthen then use of ERA as an independent data source. You could mention that.

> A: You are right, they are not assimilated. We added the suggested information: "As GNSS-based ZTD are not assimilated in ERA5 the associated ZTDs are an independent data source."

Then two comments that are more ment for eventual future work. Presumably a research vessel will carry a high quality pressure sensor. It can be expected to provide better quality ZHD than ERA on average. On top, with respect to daily variability it will be effected by the earthly and atmospheric tides as the ZHD proper, while those effects are not well represented in an NWP model such as ERA. The barometer could be used for the derivation of ZHD, to derive ZWD from the GNSS ZTD. The on-board barometer could also provide an a priori for the ZHD in the GNSS data processing, when deriving GNSS ZWD to obtain the GNSS ZTD. As humidity levels are very very low in part of the year in the Arctic, a

dominant part of the ZWD estimated in the GNSS data processing is in reality due to variability of the pressure (and hence ZHD). Using a better apriori for ZHD would reduce the problem that the mapping functions for ZHD and ZWD are not identical in the GNSS data processing.

> A: We agree with this suggestion and will incorporate the locally measured pressure data for the determination of ZHD in the future. However, at the GFZ operated GNSS sites we observed usually only very small discrepancy between local pressure measurement and ERA5 data.

Referee #2

* General comments:

This paper describes ship-borne GNSS based water vapor retrieval in the Arctic Ocean in the framework of the MOSAiC project.

After the evaluation of the GNSS raw data quality, the processing strategy is described. The solution is first assessed by studying the scattering of GNSS position estimates as the ship was in the dry dock and, next, by comparing the GNSS vertical component to tide gauge data during a harbor stay. Then, the retrieved zenith total delay are compared to ERA5, ground-based GNSS and VLBI values with conclusive results (RMS between 10 and 15 mm). Finally, IWV are derived from ship-borne ZTD and compared to radiosonde measurements (on-board launch); results are conclusive too with RMS differences around 1,5 kg/m2. Two cases of warm air intrusion are shortly described using GNSS-derived IWV.

The paper is well written, presents a relevant state-of-the-art and achieves to highlight the potentiality of the use of ship-borne GNSS antenna for ZTD/IWV retrieval for the monitoring of the atmosphere over the oceans.

The bibliography is relevant and properly formatted (some typos left, see below)

I have no major concerns but I think that some points and figures could be improved to better enhance the paper (see "Specific comments" section).

I recommend the editor to accept the papers with **minor revisions** according to the following specific comments and technical corrections.

> A: Thank you for the encouraging review and your comments.

* Specific comments & technical corrections:

- line 15: I think you could add a sentence to establish the connection between (slant) "delays" and "ZTD".

> A: The connection between slant delays and ZTD is described in line 16 and 17. To improve the clarity we replace "It is" by "The zenith delays are" in the manuscript.

- lines 85-86: please delete "therefore… estimated", estimation of ambiguity parameter is obvious (even if not fixed)

> A: We prefer to keep this sentence to indicate that all ambiguities are estimated and not only the ones which are not fixed. However, Bernese GNSS Software Version 5.2 does not support ambiguity fixing in PPP mode. We removed the advised part.

- line 86: you chose a low cutoff angle (3 deg), I guess, in order to decorrelate ZTD and height in estimation; however a low cutoff angle is more subject to multipath. Do you evaluate higher cutoff values?

> A: You are right, we selected the low cut-off to de-correlate ZTD and height but also to get as many observations as possible from the obstacle-free portside. We didn't evaluate higher cutoff values.

- Tab1: it is not clear how do you apply VMF1 for hydrostatic delay computation and mapping function. The VMF1 is position-dependent: do you update hydrostatic delay and mapping functions values with position or do you consider that due to the slow ship's move, only values for a mean position is enough? I think you could clarify this point in the text.

> A: Given the slow speed of Polarstern we assume that an averaged ship position is sufficient for the VMF computation. We added the information in table 1

- Tab1: what do you mean by "pre-eliminated" for coordinates and receiver clocks?

> A: Pre-elimination or reduction of parameters allow to reduce the normal equation system while keeping all information expect for the estimate of the pre-eliminated parameter itself. Epoch-wise parameters are pre-eliminated by default but they are be back-substituted (more details are provided in the Bernese Manual p. 183, http://www.bernese.unibe.ch/docs/DOCU52.pdf). For the coordinates we added back-substituted in the table.

- line 94: maybe you could describe shortly what is mp12.

> A: Thank you for spotting this mistake. The correct name is MP1 (L1 multipath). We corrected the text and also Fig. 3.

- Fig3: I think you should complete this figure by indicating the different periods that are mentioned between lines 94 and 106: periods when the ship is in dry dock, periods when the ship is moored, periods when water vapor measurements are not restricted.

> A: We added the dry dock, moored, and supply periods. The restricted periods are not indicated as they are not related to the tracking and processing behavior.

- Fig3: How did you explain the increase of mp12 in May 2020?

> A: Honestly, we have no explanation. The increase could be caused by re-arranged equipment in the antenna surrounding. Polarstern left the drift position mid of May which makes changes likely, i.e. additional equipment installed or stored at the observation deck.

- line 127: What is the standard deviation of the differences between antenna and tide gauge? Maybe you could add this value in the text. I guess that you took into account the vertical movements due to ocean loading, Earth tides, etc.

> A: The standard deviation is 20cm and is added to the text. Vertical movements are considered.

- line 127: How did you get the height difference between antenna and tide gauge? (empircally means: median, mean, rough draught measurement?)

> A: We used the mean difference. We added this information to the text.

- line 140: By "singular epoch", I guess you mean epoch without estimates just before and after; but what do you mean by "interpolated epochs"? Could you clarify this?

A: In case of too few observations Bernese GNSS Software used a linearly interpolation for the coordinates. We are referring here to these epochs. We modified the text accordingly.

- line 143: I am not sure to understand "ZTD estimated for epochs with fewer than 800 observations are excluded": Does this correspond to the minimum number of observations available over a 1-hour period around the time of the ZTD estimate? Could you clarify it?

A: Yes, we excluded ZTD from intervals with less than 800 observations. For clarity we replaced "epochs" by "intervals" in the text.

- line 156: Why did you use only 3-hourly ERA5 grids and not the hourly grids? Do you use the nominal ERA5 horizontal resolution (0.25° x 0.25°)?

A: We used the 0.25° x 0.25° resolution. We applied the (3-hourly) ERA5 grids for the comparison with GNSS but this comparison is done only at a 3-hourly base which is sufficient (i.e. eight values per day over 15 months) to get reliable statistics. For the later IWV computation which was done only for initial comparisons with the radiosondes we simply interpolated ZHD and T_m values. In subsequent investigations on IWV assessment, interpretation and comparison the hourly ERA5 grids could be used.

- line 161: since the ERA5 time resolution is 3h and the GNSS derived ZTD is 1h, how do you perform the outlier detection test for each ZTD values? From here, all the comparisons were realized using this "screened" dataset? (Fig8 too?)

A: The comparison was done only for the 3-hourly ZTDs. The outlier detection was applied only to determine the mean difference. Figure 8 contains all differences.

- line 168: Could you add the ground stations location on Fig 6 and, maybe, add on Fig 7 the periods when comparisons with GNSS ground stations and VLBI are done?

A: Given the map projection we added only Ny Alesund to Figure 6 but not Tromso nor Bremerhaven. We marked the intervals by adding black lines in Figure 7.

- Fig9: Could you change the colors blue/green for SAPOS/ERA5 ZTDs: it is hard to distinguish the two time series.

A: Given the good agreement between SAPOS and ERA5 the two time series are nearly identical and therefore hard to distinguish. We prefer to keep the colors as they are used similarly in Figure 10.

- line 210: Did you study the evolution of ZTD differences as distance decreases/increases?

A: We did not investigate this but it was studied for example by Wang et al., 2019. We added a corresponding sentence to the manuscript.

- line 245: Do you know which kind of radiosondes is used?

A: Vaisala RS41 radiosondes were used. For the interested reader we added the name in the text. More details are provided in the respective DOI.

- line 313: please correct doi for Boniface et al., 2012

A: corrected

- line 325: please correct doi for Fujita et al., 2008

A: corrected

- line 335: please correct doi for Hesbach et al., 2020

A: corrected

- line 374: please correct doi for Rocken et al., 2005

A: corrected

- line 394: please add doi for Zumberge et al., 1997

A: corrected

---

## Author Response (AR2)

Reply on "GNSS-based water vapor estimation and validation during the MOSAiC expedition" by Benjamin Männel et al., Atmos. Meas. Tech. Discuss., https://doi.org/10.5194/amt-2021-79-RC2, 2021

Our answers are indicated by „A:"

Editor:

Thank you for your responses to the reviewers and the modifications in the manuscript. I'm quite satisfied with most of your responses, but take into account that those responses should also be reflected in the manuscript. This latter is sometimes missing. Could you therefore provide the necessary additional information in the manuscript to the third point of reviewer 1 (more a philosophic consideration than your practical answer), and the following comments of reviewer 2: the one regarding line 86, the 2 remarks concerning Table 1, the increase of MP1 in May 2020 in Fig. 3, the remark about line 156 w.r.t. providing more details of ERA5, and the comment addressed about line 161.

Dear Roeland Van Malderen,

Thank you for your reply and for insisting on adding the answers to the manuscript which certainly helps to improve its quality. We updated the manuscript by adding the requested information. Please find a detailed list below. We highlighted the new changes in blue.

1. third point of reviewer 1 (more a philosophic consideration than your practical answer) [3 The processing is done after the expedition. Include a few sentences whether the quality of the GNSS ZTDs would be different was it done in near real-time, which is important for the potential use of ZTDs from ships in NWP.]

A: In general, near-realtime processing is beneficial and important. However, given the concept of the MOSAiC campaign this wasn't possible at all due to bandwidth limitations. Nevertheless, we added short paragraph to the processing section describing the differences (lines 111-116). As the impact of the different products should be small for ZTDs derived in a kinematic PPP for ship-based antennas we expect almost similar quality.

1. and the following comments of reviewer 2:
    a. the one regarding line 86 [line 86: you chose a low cutoff angle (3 deg), I guess, in order to decorrelate ZTD and height in estimation; however a low cutoff angle is more subject to multipath. Do you evaluate higher cutoff values?],

A: We added a statement that larger cutoff angles were not tested (lines 87-88). We rephrased the sentence accordingly as we selected the 3° cutoff angle to have access to the portside observations. In addition, using a cutoff angle of 3° is commonly used.

    b. the 2 remarks concerning Table 1 [Tab1: what do you mean by "pre-eliminated" for coordinates and receiver clocks?],

A: We added an explanation directly to the table and hope that this sentence is sufficient to describe the basic idea of parameter pre-elimination. Further information can be found e.g., Sec. 7.1.1. on pages 182-183 in Dach et al. (2015, http://www.bernese.unibe.ch/docs/DOCU52.pdf)

    c. the increase of MP1 in May 2020 in Fig. 3 [Fig3: How did you explain the increase of mp12 in May 2020?],

A: In the revised version, we did not add this information as we think, that the increase is not extraordinary nor important for the derived ZTDs. Observations causing moderate multipath are usually excluded a outliers or bad observations. Periods with large multipath are critical as discussed. However, we added our guess regarding additional equipment to the manuscript (line 107).

       d.  the remark about line 156 w.r.t. providing more details of ERA5 [line 156: Why did you use only 3-hourly ERA5 grids and not the hourly grids? Do you use the nominal ERA5 horizontal resolution (0.25° x 0.25°)?],

A: We added the explanation to the manuscript and mention that the differences were formed only for epochs with ERA5-ZTD and GNSS-ZTD (lines 169-171).

       e.  and the comment addressed about line 161 [line 161: since the ERA5 time resolution is 3h and the GNSS derived ZTD is 1h, how do you perform the outlier detection test for each ZTD values? From here, all the comparisons were realized using this "screened" dataset? (Fig8 too?)].

A: We added both information (only 3h ZTDs were used, line 170) and outliers were removed only to compute the mean (line 175).